# Use of social media in recruiting young people to mental health research: a scoping review

Megan V A Smith [ORCID],[1] Dominique Grohmann,[1] Daksha Trivedi[2]

¹Life and Medical Sciences, University of Hertfordshire, Hatfield, UK
²Health and Social Work, University of Hertfordshire, Hatfield, UK

**Correspondence to**
Megan V A Smith;
m.smith25@herts.ac.uk

## ABSTRACT

**Objectives** This review explored the literature on the use of social media in recruiting young people, aged 13–18 years, to mental health research. It aimed to identify barriers and facilitators to recruitment and strategies to improve participation in future research.

**Design** Scoping review.

**Data sources** Articles published between January 2011 and February 2023 were searched for on PubMed, Scopus, Medline (via EBSCOhost) and Cochrane Library databases.

**Eligibility criteria** Studies that outlined social media as a recruitment method and recruited participants aged 13–18 years.

**Data extraction and synthesis** Data was extracted by two reviewers independently and cross-checked by a third reviewer. Data on study design, aims, participants, recruitment methods and findings related specifically to social media as a recruitment tool were collected.

**Results** 24 journal articles met the inclusion criteria. Studies were predominantly surveys (n=13) conducted in the USA (n=16) recruiting via Facebook (n=16) and/or Instagram (n=14). Only nine of the included articles provided a summary of success and reviewed the efficacy of social media recruitment for young people in mental health research. Type of advertisement, the language used, time of day and the use of keywords were all found to be factors that may influence the success of recruitment through social media; however, as these are based on findings from a small number of studies, such potential influences require further investigation.

**Conclusion** Social media recruitment can be a successful method for recruiting young people to mental health research. Further research is needed into recruiting socioeconomically marginalised groups using this method, as well as the effectiveness of new social media platforms.

**Registration** Open Science Framework Registry (https://osf.io/mak75/).

## STRENGTHS AND LIMITATIONS OF THIS STUDY

⇒ This is the first scoping review exploring social media platforms for the recruitment of young people to mental health research.
⇒ The review used a rigorous approach to scoping and literature by following the Arksey and O'Malley's (2005) five-stage framework.
⇒ The search was conducted on studies between 2011 and 2023, and some studies that are 5–10 years old may not be relevant now due to how quickly social media platforms change in popularity.
⇒ Studies included were largely conducted in the USA, therefore results cannot be generalised to countries where internet devices and social media may not be as easily accessible or where nature of use is simply different.

## INTRODUCTION

Mental health conditions are becoming significantly more common among children and young people.[1] The impact of the COVID-19 pandemic on the mental health and well-being of children and young people has only added to the global growing concern. In 2022, over 3.5 million 6–23 year olds in England were reported to have a probable or possible mental health disorder[2] and around 14% of the world's adolescents (aged 10–19 years) are recorded to live with a mental disorder.[3] Consequently, there has been increasing demand for counselling services, hospital admissions for self-harm and referrals to specialist Child and Adolescent Mental Health Services.[4] In the UK, for example, figures have shown record numbers of referrals with the number of young people accessing mental health services reaching a new record of 708 939 in November 2022.[5] However, in the UK and globally, many countries do not have enough staff trained for dealing with mental health and across all income groups, there are reported to be just three mental health workers per 1000 000 population.[3] Adding to concern, reports do not account for the large proportion of children and young people not seeking or accessing professional for their mental health conditions.[6]

There is a dearth of evidence on the effectiveness of appropriate and timely interventions to improve the mental well-being of children and young people.[7] Consequently robust, large-scale research studies are needed to bridge the gap between supply and demand of mental health treatments.

Recruiting eligible and representative participants to research studies, however, can be challenging and is often the most time-consuming aspect of the study process.[8 9] Participation in research is generally declining, and ineffective recruitment can impact data quality and validity of research findings and lead to premature termination of the trial.[10 11] Recruitment and retention of young people for mental health research is challenging and adds another level of complexity.[10 12 13] In particular, participation may be hindered by the stigma of mental illness or fear of negative consequences from self-disclosure.[14 15]

Traditional recruitment strategies such as advertisements, telephone calls and mailing letters are often costly, time consuming and ineffective in representing the target population.[16 17] Alternatively, online recruitment strategies are now being adopted to improve enrolment outcomes.[18] One example is the use of social media, a group of mobile and internet-based applications allowing users to receive, build and share information worldwide.[19] The launch of the social media platforms such as Myspace and Facebook in 2003 and 2004, respectively, fuelled a growth of online platforms designed to increase social interconnectivity.[14] Twitter, Instagram and Snapchat, among others, shortly followed and have become an avenue for daily consumption and dissemination of information.

Most young people are active on social networking sites. In 2022, it was reported that 62% of children aged 8–17 years had profile(s) on online apps or sites, and using video-sharing platforms such as YouTube or TikTok was the most popular online activity among this age group.[20] In many cases, those with mental health problems use social media platforms to seek support networks and help others,[21] indicating that social media could be a successful method of engaging with young people for the purpose of mental health research.

An increasing number of studies are using online recruitment methods and the feasibility of such strategies is being explored. Facebook, in particular, has been shown to be a successful recruitment tool for populations who may not respond to traditional recruitment strategies, such as adolescents.[22 23] More specifically, the use of paid Facebook advertising, search tool and creation and use of a Facebook page prove successful in recruiting adolescents to health research.[22] Previous reviews that have explored social media platforms in addition to Facebook found them to be useful for recruitment to mental health research,[14 24] however these reviews did not target young people and therefore cannot be generalised to this population.

To date, and to our knowledge, a review has not explored social media platforms for the recruitment of young people to mental health research. The aim of this scoping review is to explore the literature on the use of social media in recruiting young people to mental health research studies and to identify barriers and facilitators to recruitment and strategies to improve participation.

## METHODS

Scoping reviews aim to map all of the relevant literature in a specific area of interest and consequently help to identify any gaps in existing research.[25] This approach was adopted as existing knowledge of the literature suggested a lack of previous work on how social media is used to recruit young people to mental health research.

This review uses the Arksey and O'Malley's framework[25] for scoping reviews which involves five stages: (1) identifying the research question; (2) identifying the relevant studies; (3) study selection; (4) charting the data; and (5) collating, summarising and reporting the results. This scoping review was also conducted in line with the Preferred Reporting Items for Systematic Reviews and Meta-Analysis Extension for Scoping Reviews.[26 27] The methods of this scoping review were preregistered on the Open Science Framework Registry (https://osf.io/mak75/).

### Stage 1: identifying the research questions

Four research questions directed this scoping review to address current gaps in the literature:

1. What social media platforms are described in the literature for recruiting young people for mental health research?
2. How are social media platforms used to recruit young people for mental health research?
3. What are the barriers and facilitators to recruitment of young people to mental health research using social media?
4. What are the strategies for improving recruitment and participation of young people to mental health research using social media?

### Stage 2: identifying relevant studies

On 17 February 2022, searches of the following four databases were carried out: PubMed, Scopus, Medline (via EBSCOhost) and Cochrane Library. The search was repeated on 3 February 2023 to identify relevant papers published since February 2022. Medical subject headings were searched using Boolean operators 'OR/AND'. The search terms were: (adolescent OR teenager OR youth OR 'young adult') AND ('mental health' OR 'mental illness*' OR 'mental disorder') AND ('social media' OR 'social network*') AND (recruit* OR advert*). The search format used in the databases was modified to meet their requirements. Further details regarding the search terms used are provided in table 1. The final search strategy for databases can be found in online supplemental file 1.

As past reviews in this area have often focused on the use of Facebook,[18 22] this review aimed to capture publications exploring all social media platforms developed since. As Instagram saw significant growth in December 2010, articles published prior to 1 January 2011 were not included in the database searches, allowing time for any use of Instagram for recruitment to be reported. Therefore, the review years spanned from January 2011 to February 2023.

**Table 1** Search terms

| Search term 1 | Search term 2 | Search term 3 | Search term 4 |
|---|---|---|---|
| Search operator | AND | AND | AND |
| Adolescent | 'Mental health' | 'Social media' | Recruit* |
| Adolescent | Mental Health | Social media | Advert* |
| Teenager | 'Mental illness*' | 'Social network*' | Advertising |
| Youth | Mental disorders | Social Networking | |
| 'Young adult' | Mental disorder | | |
| Young adult | | | |

Medical subject headings (MeSH) terms (MeSH major topic) are highlighted in green.

### Stage 3: study selection

Studies were included if they (a) outlined the use of social media (eg, Twitter, Instagram, Facebook, Snapchat and TikTok) as a method to recruit young people aged 13–18 years and/or those with a mean age of 13–18 years, (b) addressed a mental ill-health (eg, depression, anxiety or eating disorder) and (c) were published between January 2011 and February 2023. Studies were excluded from this review if they met any of the following criteria:

1. Did not refer to the use of social media for recruitment of young people aged 13–18 years in the methodology.
2. Were not written in the English language (due to the time and cost involved in translating them to English).
3. Did not address a mental ill-health.

Retrieved records from the database searches were extracted and imported into Rayyan, a free web tool created to facilitate literature reviews.[28] One reviewer (MVAS) scanned all imported records and removed duplicates. Titles and abstracts were screened to define the eligibility of each article by two researchers (MVAS, DG) independently following the aforementioned predefined selection criteria. Publications with a title or abstract not meeting the eligibility criteria were excluded. Following the initial screening, full-text articles were reviewed by both researchers independently to make a final decision of inclusion. Any discrepancies regarding the inclusion of articles were resolved by discussion and mutual agreement with the third reviewer (DT). The full text of relevant papers was retrieved for further analysis by two reviewers (MVAS, DG) and was either included or excluded for review based on the eligibility criteria presented below. Primary studies and/or relevant systematic reviews that met the inclusion criteria were included. However, due to the broad age range of two relevant systematic reviews, included articles were individually screened for eligibility. From this, one article met inclusion criteria for the current scoping review and was consequently included in analysis.

The reference lists of all included publications were also examined to ensure all relevant and eligible resources had been identified.

### Stage 4: charting the data

A data extraction form was developed in Excel by one of the reviewers and agreed by all to determine which variables to extract. Relevant articles were charted using the following column headings:

1. Author(s), year of publication.
2. Study design.
3. Study aim(s).
4. Participants.
5. Recruitment method(s).
6. Social media platforms.
7. Recruitment related findings.

The two reviewers independently charted the data, discussed the results and continuously updated the data-charting form in an iterative process.

### Stage 5: collating, summarising and reporting the results

All extracted data from the included articles were summarised and tabulated by a member of the research team. In line with Arksey and O'Malley,[25] the narrative account is presented in two ways. First, the nature and distribution of the studies included, for example, study design, country, settings and participants group, were reported. Second, the literature was organised according to the following themes which were drawn from the research questions: methods of social media recruitment to recruit young people to mental health research, barriers and facilitators to recruitment of young people to mental health research using social media and how to improve recruitment and participation of young people to mental health research using social media.

### Patient and public involvement

There was no patient and public involvement due to the nature of this research.

### RESULTS

### Included studies

The searches in 2022 and 2023 are presented together as the same search methodology was adopted. Initial searching of the electronic databases provided 3669 records. In total, 402 duplicates were removed leaving

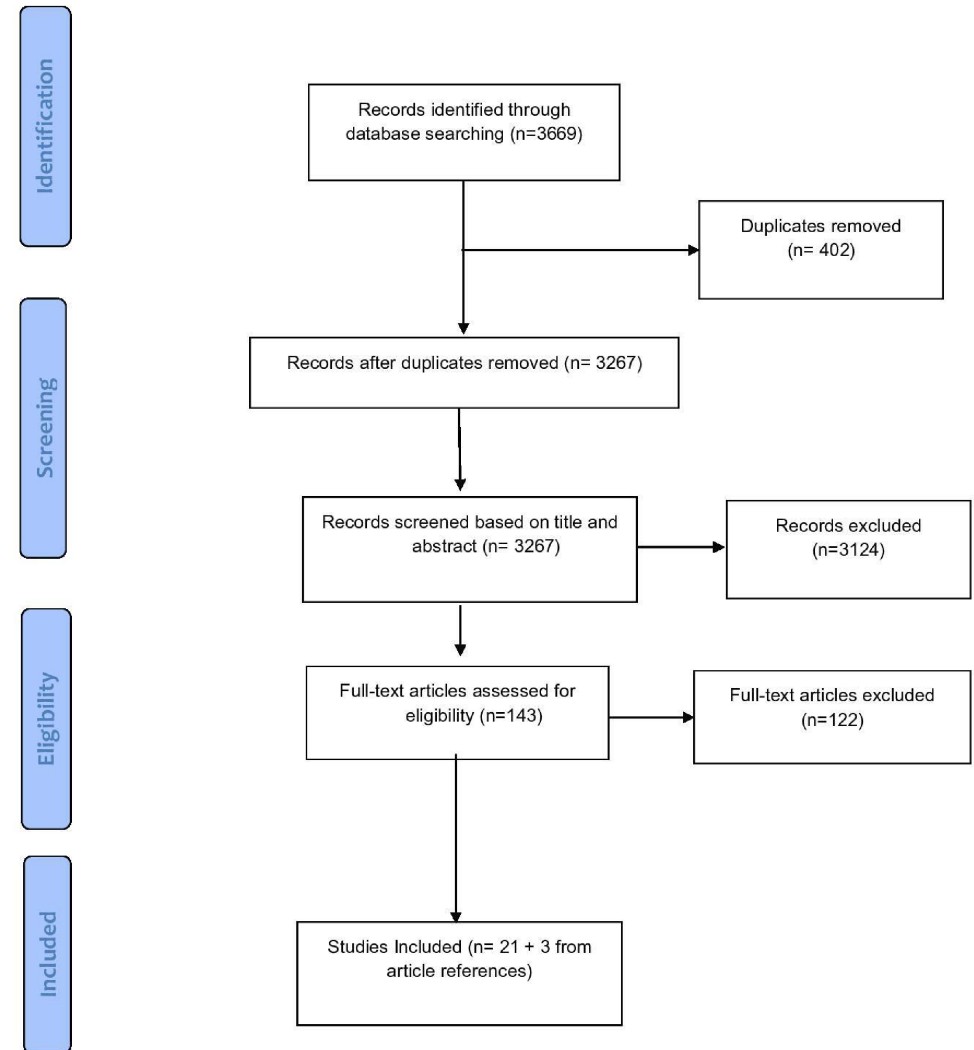

**Figure 1** Preferred Reporting Items for Systematic Reviews and Meta-Analysis flow diagram.

3267 records to screen for relevance based on title and abstract. Based on title and abstract screening, 3124 were excluded. In total, 143 full-text articles were assessed by the researchers for inclusion. In total, 122 were excluded. The majority of articles were excluded due to not addressing mental ill-health or not including the correct sample, that is, young people aged 13–18 years. Other reasons for exclusion included incorrect study design or recruitment technique. A total of 21 studies met all inclusion criteria and were subsequently included within this scoping review. When screening the references of the 21 included articles, 2 articles[29 30] were found to fit the inclusion criteria and were therefore also included within the review. As mentioned previously, one paper[31] was included following the screening of two potentially eligible systematic reviews. An overview of the study selection process is provided in figure 1.

### Study characteristics
Characteristics of the included articles are summarised in online supplemental file 2. Study designs included 13 surveys,[29–41] 3 pilot randomised controlled trials,[42–44] 4

feasibility studies,[45–48] 2 randomised controlled trials[49 50] and 1 mixed methods exploratory study.[51] One paper[52] presents the protocol of a longitudinal study using a subset of Goldbach et al's[38] sample. Schrager et al[52] provide detailed information of the recruitment methods used by Goldbach et al.[38] For the purpose of this review, they are presented as separate studies.

Of the included articles, 16 were conducted in the USA,[29 33 37–48 50 52] 3 in Canada,[31 34 35] 1 in Australia,[51] 1 in Sweden[49] and 1 in Brazil.[53] One of the included articles recruited participants from both the USA and Brazil[30] and one recruited participants from 'major English-speaking countries' including Australia, the USA, the UK and New Zealand.[32]

### Participants
Some articles grouped their entire sample by age,[33 34 37 41] therefore only the data from the groups that were eligible for this review were considered (13–18 years olds). Four studies recruited youth with a mental health condition[32 45 47 49] and two recruited athletes.[39 40] Nine studies recruited adolescents from specific demographic

backgrounds, for example, sexual and gender minorities,[29 34 38 44 48 52] America Indian and Alaska Native youth[50] and heterosexually active black youth.[42] Four studies recruited youth based on social media activity, that is, posting online about being sad or depressed,[33 43] females engaging in pro-eating disorder posts[37] and individuals who had liked online social media pages for the television series '13 Reasons Why'.[30 36] One study recruited youth who had been affected by violence[31] and one recruited youth seeking support for issues related to family discord and associated impacts on emotional well-being.[51] The remaining three studies recruited youth with no particular characteristics noted.[41 46 54]

## Methods of social media recruitment

Among all retrieved articles, 20 described at least one specific type of social media as a method of recruitment (see online supplemental file 2 for distribution). Facebook was used most commonly by 16 of the 24 included articles,[30–33 35–41 45 48 50–52] 14 used Instagram[29 33 35 37 38 41 44–48 50–52] and 5 used Twitter.[32 33 37 39 40] Other, less commonly used social media platforms were Reddit,[33 37 45] Tumblr,[30 33 43] Snapchat,[45 51] TikTok[45] and YouTube.[38 45 52] Three articles, although discussing the use of social media for recruitment, did not specify the platforms used.[34 42 49]

Of the articles which specified the platforms used, 12 of these used more than one social media platform to advertise.[29 32 33 37–41 45 48 50 52] Half of the articles (n=12) used advertisements tailored to reach their target population by setting restrictions, for example, keywords, age, interests or location.[29 30 32 33 37 38 41 45 46 48 50 52] Three articles specified using Facebook Business Manager to manage Instagram and/or Facebook advertisements.[45 46 50]

Some articles used a combination of online and offline advertisement methods. Brawner et al,[42] for example, initially aimed to recruit through community-based mental health providers, high schools, community partners (eg, recreation centres) and provider referrals; however, this was not successful, so they introduced social media study promotion. Lattie et al[47] advertised via Instagram as well as offline, for example, through schools, community settings, fliers, university databases and at conferences. Morgan et al[32] also used a combination of paid advertisements on Facebook and Google as well as hard copy advertisements in youth mental health clinics and support groups. Amon et al[51] recruited specifically through the Kids Helpline (KHL) website, referrals from KHL counsellors and schools, as well as through social media. Goldbach et al[38] and Schrager et al[52] used social media advertising along with respondent-driven sampling, where participants could earn gift cards through referring eligible participants. Mechler et al[49] also recruited through contacts with schools, youth associations, social workers and healthcare providers.

## Barriers to social media recruitment

Few articles discussed barriers to social media recruitment. One potential barrier highlighted was that this method of recruitment may be more biased to individuals with a higher socioeconomic status who have access to internet facilities and therefore likely to be active on social media[31 35 39 40] potentially leading to an unrepresentative sample. Cost may be seen as a potential barrier, and several studies revealed their total cost of using social media advertising: US$1351[31]; US$1536[48]; US$1591[46]; and over US$5000.[50] One study highlighted the potential complexity of using social media advertising, stating that using Facebook Business Manager to monitor their advert statistics could be complicated.[46]

## Facilitators of social media recruitment

In total, 6 of the 24 included articles discussed facilitators of social media recruitment in detail. Regarding most effective type of social media platform for recruitment, Kasson et al[45] found Snapchat to be the most successful, with Instagram as the second most successful in a comparison of five social media sites. Fitzsimmons-Craft et al[37] also found Instagram to be highly effective at recruiting their adolescent age group (15–17 years olds) with 76% of the sample recruited through this platform. Kutok et al's study[46] compared combinations of Instagram advertisements and found that those with the campaign strategy 'Traffic' (targeting users that often click on links within an ad) and the advertisement placement 'Feed' (on Instagram's regular feed of posts, as opposed to a 'Story' which disappears after 24 hours) were the most effective combination in terms of recruitment success and cost-effectiveness. Regarding wording of social media advertisements, two studies found that those with a more positive tone or those that suggested helping the researchers received more interest.[46 50] Chu and Snider[31] investigated Facebook advertising throughout the week and found the number of clicks and impressions were much higher on weekends than weekdays. One study by Kelleher et al[43] successfully recruited youth through Tumblr who had posted about depression, through searching #depress to identify potentially suitable individuals. Several studies made use of 'labels' or 'keywords' in order to target a certain population,[37 45 48] however the effectiveness of this strategy in aiding recruitment is difficult to determine as there were no comparator advertisements.

## How to improve recruitment and participation of young people to mental health research using social media

Few articles addressed methods to improve recruitment of this population group using social media. Several studies found that advertising through Instagram was beneficial in their recruitment, which suggests that it may be a suitable platform in attracting young people to mental health research.[37 45] Snapchat was also found to be highly effective at recruiting.[45] Both of these social media platforms are heavily image based which may have led to the increase in interest and is a factor to consider in future

research. Kutok et al[46] noted that social media is an ever-changing environment and therefore highlighted the importance of being flexible. For example, Instagram and Snapchat may be effective currently, however in several years' time other platforms may be more popular and therefore researchers should be open to trying a range of platforms. Several studies found that adverts with a more positive tone received more interest,[46 50] which suggests that the language used when advertising the study should be an important consideration for future studies.

## DISCUSSION
### Main findings
The aim of this scoping review was to explore the literature on the use of social media in recruiting young people to mental health research and to identify barriers and facilitators to recruitment and strategies for improving recruitment. From the 3308 articles identified through the four databases, 24 studies met the inclusion criteria.

The included studies investigated a range of populations recruited successfully, suggesting that social media may be a suitable method for engaging with and recruiting specific populations. This is consistent with previous reviews investigating social media as a recruitment method for health research.[55 56] However, further research is required to determine the suitability of digital delivery to socioeconomically and digitally marginalised youth to mental health research.[57] This is even more crucial considering socioeconomically marginalised, and consequently digitally excluded, youths are more likely to develop mental health problems than more socioeconomically advantaged peers.[57–59]

Regarding type of social media platform, the majority of studies used the platform Facebook, consistent with previous reviews focusing on the general population.[22 23] Facebook use in 2014–2015 among teens was at 71%, which has reduced considerably to 32% in 2021.[60] This decrease in popularity could imply that other social media platforms may be preferred when targeting this population and highlights the importance of keeping up to date with the most popular platforms with this age group at the time of recruitment. Instagram was also a popular platform of choice. Kasson et al[45] found Snapchat and Instagram to be the most effective social media sites regarding percentage of the sample recruited. They suggested that the nature of these sites being heavily image based may attract this population compared with other networking sites. However, few of the studies compared effectiveness across a range of platforms, therefore these findings should be interpreted with caution and comparison of sites is something that should be investigated further in the future.

Regarding barriers to using social media as a recruitment technique, one of the main concerns is the potential cost of advertising, with included studies ranging from US$1351 to over US$5000. Affordability will largely depend on decisions made by the research team and the funding that has been allocated, however this highlights a need for future studies to investigate cost-effectiveness of social media compared with other forms of recruitment to determine which is more appropriate, particularly if teams have a limited budget. Another potential barrier to using social media for recruitment is the potential complexity of advertising sites, however this is likely to vary depending on the research team's knowledge and experience in this area. An additional barrier and potential consideration are the caveats of minimum age requirements to sign up to social networking sites, as well as specific considerations around consent and confidentiality. This must not be overlooked if social media are to be used in adolescent research.[22]

### Strengths and limitations
To our knowledge, this is the first scoping review exploring all social media platforms for the recruitment of young people to mental health research. Furthermore, it followed the rigorous Arksey and O'Malley's[25] five-stage framework which allowed for transparency. To appreciate the findings of this review the following limitations should be acknowledged. First, not all published articles will necessarily discuss recruitment in the title or abstract and therefore some articles may have been overlooked by the search. It is important to also note that the search was conducted from 2011 to 2023, and due to the ever-changing nature of popularity of social media sites, some of the studies that were conducted several years ago may not be as relevant now. Another limitation is that all included studies were from Western countries such as Canada and the USA, therefore the results cannot be generalised to countries in which social media use is restricted or different from these countries. Some of the social media platforms investigated here are banned in countries including Iran, China and Uganda, therefore results relating to certain platforms may not be applicable. Finally, only a few of the papers included reported data on the specifics of using social media as a recruitment method, for example, the days that are best to advertise, tone of advert and any keywords used, therefore these findings should be interpreted with caution.

### Future suggestions
Although the findings and reflections from these papers may be of help for other researchers who are looking to recruit using social media, many of those included did not evaluate the effectiveness of social media as a recruitment technique. Therefore, we suggest that future studies provide more detail and report on the effectiveness of various strategies adopted such as the platform, type of advertisement and language used, as well as the number of participants recruited through social media compared with more traditional recruitment methods if these are also adopted. As this method of recruitment increases in popularity, a systematic review could be conducted to include a more comprehensive search to ensure those publications not mentioning recruitment

methods in their title and abstract are not disregarded from the search. Additionally, it would be beneficial for researchers to investigate the cost-effectiveness of social media compared with other types of recruitment to ascertain whether it is a cost-effective method. Such findings may then contribute towards developing a guidance document on this topic that can be shared among researchers to increase knowledge and awareness when considering this recruitment method for their research. As mentioned, the popularity of social media platforms can change over time and consequently so can the effectiveness of their use in recruitment. Therefore, future research could look to explore recruitment trends across different platforms, countries and mental health conditions longitudinally.

## CONCLUSIONS

Recruiting young people to mental health research can be challenging and, increasingly, online recruitment methods are being used to increase enrolment and accessibility to participants. This review concludes that social media can be a successful method for recruiting young people to mental health research in terms of reaching the target number of participants, recruiting specific populations and in some cases be more successful compared with traditional methods of recruitment. Further research is needed into recruiting socioeconomically marginalised groups using this method, as well as the effectiveness of new social media platforms. As technology continues to advance, so must recruitment methods.

**Acknowledgements** The authors would like to thank Lauren Denyer for her assistance in developing the protocol for this scoping review.

**Contributors** MVAS and DG contributed to the conception and design of this review. MVAS conducted the search and selection process. MVAS and DG extracted all data. DT provided advice and guidance on the analysis and interpretation of results. MVAS and DG produced the first draft of the manuscript. All authors contributed to writing and approved the final draft of the manuscript. MVAS acts as guarantor and accepts full responsibility for the finished work and/or the conduct of the study, had access to the data and controlled the decision to publish.

**Funding** This work was supported by the Health Technology Assessment funding stream of the National Institutes of Health Research (Reference Number: 17/78/10).

**Competing interests** None declared.

**Patient and public involvement** Patients and/or the public were not involved in the design, or conduct, or reporting, or dissemination plans of this research.

**Patient consent for publication** Not applicable.

**Ethics approval** Not applicable.

**Provenance and peer review** Not commissioned; externally peer reviewed.

**Data availability statement** Data sharing not applicable as no datasets generated and/or analysed for this study.

**ORCID iD**
Megan V A Smith http://orcid.org/0000-0002-1482-2350

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
