## [Reviewer comments · BMJ Open]

ARTICLE DETAILS

TITLE (PROVISIONAL)	The use of social media in recruiting young people to mental health research: a scoping review
AUTHORS	Smith, Megan; Grohmann, Dominique; Trivedi, Daksha

VERSION 1 – REVIEW

REVIEWER	Katherine Cohen Stony Brook University
REVIEW RETURNED	10-Jun-2023

GENERAL COMMENTS	Thank you for the opportunity to review this interesting paper that reviews papers using social media to recruit young people in mental health research. The following comments are intended to help strengthen this manuscript. I hope they are helpful for the authors. 1. A concern I have is that many papers don't mention their recruitment methods in the title or abstract. Therefore, I'm afraid that many papers may not have been included in the review that would fit the criteria. I believe a more comprehensive search could be conducted to determine if there are additional articles that fit the inclusion criteria. For example, the search terms included social media or social network. It is possible that more articles may have been found if the search terms included names of specific social media platforms.2. It may be helpful to include a reference for the first sentence in the introduction.3. The first paragraph of the introduction focuses solely on statistics from the UK. However, the scoping review includes articles from any country. It may be useful to include more global statistics in the introduction.4. The article would benefit from including more information regarding the studies' success in recruitment on social media. For example, how many people did the ads reach? How many people clicked the ads? How many people enrolled in the studies after clicking the ads? What were the dropout rates in the studies?5. The social media sites that this study focuses on are common in the West, but I am wondering whether social media sites used more in the East such as WeChat should be included.6. The PRISMA Flow Diagram would benefit by including the specific reasons why records were excluded.7. It is not clear to me if or how interrater reliability was calculated.8. The major barrier I hear about as a researcher who uses social media for recruitment in mental health trials is bots. Did any of the articles mention this problem?
--

REVIEWER	J Broughan University College Dublin, School of Medicine
REVIEW RETURNED	18-Sep-2023

GENERAL COMMENTS	Comments to the Author: This is an excellent manuscript. It's an important topic as social media is used so much for recruitment and this trend will surely increase. The rationale is communicated very well, the methods are applied very well, the results are clear and informative, and so is the discussion. I added some more detailed feedback below but it is quite minor. Happy to recommend for publication if these minor issues are addressed. 1. Abstract>Data sources: "...databases were searched from January 2011 to February 2023." Small thing but this reads like you searched for papers for 12 years. Suggest saying something like "Articles published between January 2011 and February 2023 were searched for on PubMed..." 2. Introduction section is excellent. 3. Methods>Page 5>line 45: As past reviews in this area have often focused on the use of Facebook (e.g. (16,20) Close the bracket (e.g. (16,20)) 4. Methods>Page 6>line 13: "Were an animal study" Don't think you need to specify this. 5. Methods>Page 6>lines 22-23: "Following the initial screening, full-text articles were reviewed to make a final decision of inclusion." Please state whether these were screened by one or two reviewers. 6. Table 2>Amon et al.>Study aim(s): Please use the unabbreviated version of SNS. 7. Results>Barriers to social media recruitment>page 17>line7: One potential barrier highlighted was that this method of recruitment may be more bias to individuals with a Suggest saying "...may be more biased to individuals with..." 8. Results>Facilitators of social media recruitment "(43) found Snapchat to be the most successful... The approach of using the reference number in this way rather than the author's name followed by the reference (e.g., Smith et al (43)) number seems unusual. I don't mind either way and if the journal is happy with this approach I am too. Suggest checking with them. Just thought I should mention it. There are a number of examples of this. 9. Discussion>Future suggestions> It struck me that the quality of recruitment can change over time on platforms. For instance, you said Facebook's effectiveness for recruitment declined. Perhaps another suggestion for future research would be a longitudinal study(ies) that track recruitment trends over time across different platforms, countries, cultures, etc?
--

VERSION 1 – AUTHOR RESPONSE

Reviewer: 1
Dr. Katherine Cohen, Stony Brook University

Comments to the Author:
Thank you for the opportunity to review this interesting paper that reviews papers using social media to recruit young people in mental health research. The following comments are intended to help strengthen this manuscript. I hope they are helpful for the authors.

Response: We would like to thank the reviewer for taking the time to review our manuscript and for providing helpful feedback.

1. A concern I have is that many papers don't mention their recruitment methods in the title or abstract. Therefore, I'm afraid that many papers may not have been included in the review that would fit the criteria. I believe a more comprehensive search could be conducted to determine if there are additional articles that fit the inclusion criteria. For example, the search terms included social media or social network. It is possible that more articles may have been found if the search terms included names of specific social media platforms.

Response: We thank you for your comment. In our experience, although papers may specify the social media platform used i.e. Facebook, they usually have "social media" or "social network" as a key or MeSH term and our search included such publications. However, we understand your concern and in the aid of transparency, have included your comment within our limitation section "*not all published articles will necessarily discuss recruitment in the title or abstract and therefore some articles may have been overlooked by the search*". We also suggest that this method may be used in a systematic review in the future as this method of recruitment increases in popularity within the Future suggestions section "*As this method of recruitment increases in popularity, a systematic review could be conducted to include a more comprehensive search to ensure those publications not mentioning recruitment methods in their title and abstract are not disregarded from the search.*"

2. It may be helpful to include a reference for the first sentence in the introduction.

Response: Thank you for your suggestion, we have added a reference which discusses the increasing global burden of mental health problems among children and adolescents - Hossain, M. M., Nesa, F., Das, J., Aggad, R., Tasnim, S., Bairwa, M., ... & Ramirez, G. (2022). Global burden of mental health problems among children and adolescents during COVID-19 pandemic: An umbrella review. *Psychiatry Research*, 114814.

3. The first paragraph of the introduction focuses solely on statistics from the UK. However, the scoping review includes articles from any country. It may be useful to include more global statistics in the introduction.

Response: Thank you for highlighting the absence of global evidence in this area. We have restructured the first paragraph and included references from a global perspective to emphasise the impact of mental health on adolescents worldwide, not just from a UK perspective. The first paragraph now reads:

"Mental health conditions are becoming significantly more common among children and young people (1). The impact of the COVID-19 pandemic on the mental health and wellbeing of children and young people has only added to the global growing concern. In 2022, over 3.5 million 6-23 year olds in England were reported to have a probable or possible mental health disorder (2) and globally, around 14% of the world's adolescents (aged 10-19 years) are recorded to live with a mental disorder (3). Consequently, there has been increasing demand for counselling services, hospital admissions for self-harm and referrals to specialist Child and Adolescent Mental Health Services (CAMHS) (4). In the UK, for example, figures have shown record numbers of referrals with the number of young people accessing mental health services reaching a new record of 708,939 in November 2022 (5). However, in the UK, and globally, many countries do not have enough staff trained for dealing with mental health and across all income groups, there are reported to be just three mental health workers per 1000 000 population (3). Adding to concern, reports do not account for the large proportion of children and young people not seeking or accessing professional for their mental health conditions (6)."

4. The article would benefit from including more information regarding the studies' success in recruitment on social media. For example, how many people did the ads reach? How many

people clicked the ads? How many people enrolled in the studies after clicking the ads? What were the dropout rates in the studies?

Response: We thank you for your helpful feedback and suggestion for the inclusion of more information regarding the studies' success in recruitment on social media. Unfortunately, as mentioned in the manuscript, the included studies rarely discussed social media recruitment in detail and therefore we do not have the information suggested. Where the detail was presented in the research articles, we have included that information in our manuscript and summarised in table 2 in the column titled "recruitment related findings" .

5. The social media sites that this study focuses on are common in the West, but I am wondering whether social media sites used more in the East such as WeChat should be included.

Response: We thank you for your suggestion. Our search was not explicit to specific social media platforms and therefore we did not restrict our search to only focus on social media sites common in the West. In discussing the included articles, we presented all social media sites used. As you can see from Table 2, the included articles used the social media platforms Facebook, Instagram, Snapchat, Twitter, Reddit, TikTok, Tumblr and YouTube. In our limitation section, we discuss that all included studies were from Western countries and therefore the results cannot be generalised to other countries.

6. The PRISMA Flow Diagram would benefit by including the specific reasons why records were excluded.

Response: We thank you for your suggestion. We used the online review tool Rayyan to screen all articles and unfortunately this tool does not have the option to differentiate reason for exclusion by point of screening i.e. whether they were excluded during the screening based on title and abstract or whether they were excluded when screening the full text. We therefore do not think it's appropriate to add into the PRISMA Flow Diagram as we cannot provide the accurate data. We have, however, added information in the text under 3.1. Included studies to provide an overall summary of reasons for exclusion – *"The majority of articles were excluded due to not addressing mental-ill health or not including the correct sample i.e., young people aged 13-18. Other reasons for exclusion included incorrect study design or recruitment technique"*.

7. It is not clear to me if or how interrater reliability was calculated.

Response: Interrater reliability calculation is not a requirement of scoping reviews, and usually found in systematic reviews. However, we have specified in the study selection section that any discrepancies regarding the inclusion of articles were resolved by discussion and mutual agreement with the third reviewer. We have clarified in the text in section 2.3. Stage 3: Study selection to show that any disagreements were resolved with a third reviewer *"Following the initial screening, full-text articles were reviewed by both researchers independently to make a final decision of inclusion. Any discrepancies regarding the inclusion of articles were resolved by discussion and mutual agreement with the third reviewer (DT)"*.

8. The major barrier I hear about as a researcher who uses social media for recruitment in mental health trials is bots. Did any of the articles mention this problem?

Response: Thank you for your insight and query. Unfortunately, none of the articles selected for inclusion in our manuscript mentioned bots. However, if this is an ongoing issue in mental health trials, this may be something of interest for future research.

Reviewer: 2
Dr. J Broughan, University College Dublin

Comments to the Author:

This is an excellent manuscript. It's an important topic as social media is used so much for recruitment and this trend will surely increase. The rationale is communicated very well, the methods are applied very well, the results are clear and informative, and so is the discussion. I added some more detailed feedback below but it is quite minor. Happy to recommend for publication if these minor issues are addressed.

Response: We would like to thank this reviewer for taking the time to review our manuscript and for their supportive comments. We have taken all feedback on board and as you will see from the response to the comments, have made the appropriate amendments.

1. Abstract>Data sources: "...databases were searched from January 2011 to February 2023." Small thing but this reads like you searched for papers for 12 years. Suggest saying something like "Articles published between January 2011 and February 2023 were searched for on PubMed..."

Response: Thank you for making this suggested edit to the text, we can see how this can read have amended the text in line with your suggestion so it now reads "*Articles published between January 2011 and February 2023 were searched for.....*"

2. Introduction section is excellent.

Response: Thank you for your kind comment regarding the introduction.

3. Methods>Page 5>line 45: As past reviews in this area have often focused on the use of Facebook (e.g. (16,20) Close the bracket (e.g. (16,20))

Response: Thank you for highlighting the missing bracket, we have now closed the bracket.

4. Methods>Page 6>line 13: "Were an animal study" Don't think you need to specify this.

Response: Thank you for this suggestion, we agree this is not necessary as our inclusion criteria specifies that studies are required to recruit young people. In agreement, we have removed "were an animal study" from the exclusion criteria.

5. Methods>Page 6>lines 22-23: "Following the initial screening, full-text articles were reviewed to make a final decision of inclusion." Please state whether these were screened by one or two reviewers.

Response: Thank you for your comment asking for clarification regarding the screening of full-text articles. The full-text articles were reviewed by two reviewers and the text now reads "*Following the initial screening, full-text articles were reviewed by both researchers independently to make a final decision of inclusion*" for clarity.

6. Table 2>Amon et al.>Study aim(s): Please use the unabbreviated version of SNS.

Response: Thank you for highlighting this, we have now unabbreviated SNS to "*social networking service*"

7. Results>Barriers to social media recruitment>page 17>line7: One potential barrier highlighted was that this method of recruitment may be more bias to individuals with a Suggest saying "...may be more biased to individuals with..."

Response: Thank you for your recommendation, we have amended the text to reflect your suggestion.

8. Results>Facilitators of social media recruitment "(43) found Snapchat to be the most successful..."

The approach of using the reference number in this way rather than the author's name followed by the reference (e.g., Smith et al (43)) number seems unusual. I don't mind either way and if the journal is happy with this approach I am too. Suggest checking with them. Just thought I should mention it. There are a number of examples of this.

Response: We thank you for highlighting this error and we think it may be due to the referencing manager we used automatically changing references. We have read through the manuscript and edit the references to reflect your suggestion of the author's name appearing before the reference and hope this now reads clearly.

9. Discussion>Future suggestions>

It struck me that the quality of recruitment can change over time on platforms. For instance, you said Facebook's effectiveness for recruitment declined. Perhaps another suggestion for future research would be a longitudinal study(ies) that track recruitment trends over time across different platforms, countries, cultures, etc?

Response: Thank you for this helpful insight and suggestion. We have added this suggestion for future research to the end of the Future suggestions section and it now reads *"As mentioned, the popularity of social media platforms can change over time and consequently so can the effectiveness of their use in recruitment. Therefore, future research could look to explore recruitment trends across different platforms, countries and mental health conditions longitudinally."*